# Horticultural Image Feature Matching Algorithm Based on Improved ORB and LK Optical Flow

Qinhan Chen, Lijian Yao *, Lijun Xu, Yankun Yang, Taotao Xu, Yuncong Yang and Yu Liu

College of Optical, Mechanical and Electrical Engineering, Zhejiang A&F University, Hangzhou 311300, China
* Correspondence: ljyao@zafu.edu.cn

**Abstract:** To solve the low accuracy of image feature matching in horticultural robot visual navigation, an innovative and effective image feature matching algorithm was proposed combining the improved Oriented FAST and Rotated BRIEF (ORB) and Lucas–Kanade (LK) optical flow algorithm. First, image feature points were extracted according to the adaptive threshold calculated using the Michelson contrast. Then, the extracted feature points were uniformed by the quadtree structure, which can reduce the calculated amount of feature matching, and the uniform ORB feature points were roughly matched to estimate the position of the feature points in the matched image using the improved LK optical flow. Finally, the Hamming distance between rough matching points was calculated for precise matching. Feature extraction and matching experiments were performed in four typical scenes: normal light, low light, high texture, and low texture. Compared with the traditional algorithm, the uniformity and accuracy of the feature points extracted by the proposed algorithm were enhanced by 0.22 and 50.47%, respectively. Meanwhile, the results revealed that the matching accuracy of the proposed algorithm increased by 14.59%, whereas the matching time and total time decreased by 39.18% and 44.79%, respectively. The proposed algorithm shows great potential for application in the visual simultaneous localization and mapping (V-SLAM) of horticultural robots to achieve higher accuracy of real-time positioning and map construction.

**Keywords:** feature matching algorithm; improved ORB algorithm; optical flow method; horticultural image; horticultural robot

## 1. Introduction

Visual simultaneous localization and mapping (V-SLAM) technology is critical for the visual navigation of horticultural robots [1,2]. However, owing to the poor uniformity of feature point extraction and low matching accuracy of environmental images caused by complex textures and similar feature information, its accuracy in real-time positioning and scene reconstruction can be severely impeded [3]. Therefore, many studies have been conducted on the optimization of feature matching. Generally, this type of research can be divided into feature extraction and feature matching [4].

For feature extraction, Rublee et al. proposed a directional binary simple description (i.e., Oriented FAST and Rotated BRIEF (ORB)) algorithm that significantly improves the speed of feature extraction [5]. However, the image feature points extracted by this algorithm are concentrated and do not exhibit scale invariance. Integrating scale-invariant feature transform (SIFT) features with ORB features can effectively improve the scale invariance and quality of the feature points [6,7]. However, this method increases the time consumption of feature extraction. Xu et al. utilized an octagon filter bank (DFOB) to extract feature points [8]. Cai et al. proposed an ORB method based on affine transformation [9]. Both algorithms contribute to improving the number and speed of feature point extraction. The drawbacks of these algorithms include redundant feature points and additional time required for feature matching.

The purpose of feature matching is to find sufficient and accurate correspondences from two or more overlapped images and a variety of studies have been conducted [10]. Zhang et al. presented a coarse-to-fine, large-size, high-resolution image registration method for feature matching [11]. This approach uses a compute unified device architecture (CUDA) to speed up image matching, which improves the speed of feature matching but requires additional computing equipment. Shi et al. designed an accelerated matching algorithm using network topology [12]. This algorithm exhibits poor robustness when the feature points are repetitive. Chen et al. proposed a new low-complexity image-matching algorithm that uses a local multi-feature hashing (LMFH) descriptor to simplify feature comparison [13] to improve the efficiency of feature matching, but its performance is poor in environments with a large number of dense features. Pang et al. presented an image feature matching algorithm based on a weak supervised learning method using graph convolutional MLPs and Siamese neural networks on unstructured geometric feature points [14]. This algorithm improves the accuracy and robustness of feature matching, but requires large amounts of data to train the model; therefore, it is not universal.

Traditional feature matching comprises three phases: feature extraction, feature point description, and feature vector matching [15]. Usually, the random sample consensus algorithm, relaxation iteration method, minimum median method and parallax-based filtering algorithm are required to eliminate mismatches, which also reduces the real-time performance of the feature-matching algorithm [16–18]. The feature matching algorithm based on the optical flow technique can improve the efficiency of calculation speed and high frequency [19]; however, it needs to meet the strong assumption of invariability of grayscale, thus lacking robustness in practical applications.

In this study, we propose an innovative and effective horticultural image feature matching algorithm based on improved ORB and LK optical flow techniques. The experimental results reveal that the proposed algorithm performs better than traditional image feature matching techniques for various parameters. A significant increase in the uniformity of feature points, accuracy, and robustness was observed in horticultural image feature matching. This makes it suitable for horticultural robot navigation, which requires stability and accuracy of real-time positioning and scene reconstruction.

## 2. Methodology

### 2.1. Algorithm Framework

As shown in Figure 1, the algorithm structure in this study consisted of two parts: improved ORB feature point extraction and combined feature matching.

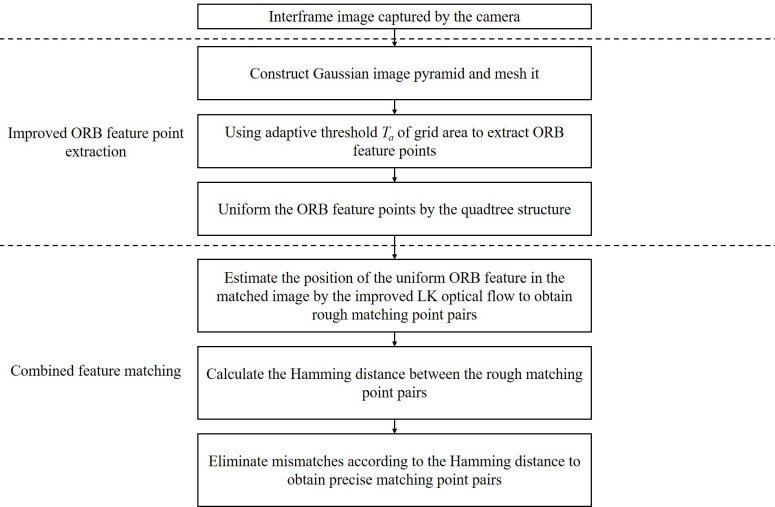

**Figure 1.** Proposed algorithm framework.

### 2.2. Improved ORB Feature Point Extraction

#### 2.2.1. Construct Gaussian Image Pyramid

The original image is the bottom layer (i.e., the 0th layer) of the Gaussian pyramid (Figure 2). Every time the image moves to the upper layer, Gaussian filtering and fixed magnification reduction are performed, and an image pyramid with an ascending resolution from high to low can be obtained, as shown in Figure 3. In feature matching, scale invariance is achieved by matching the images of different layers of the image pyramid at adjacent times.

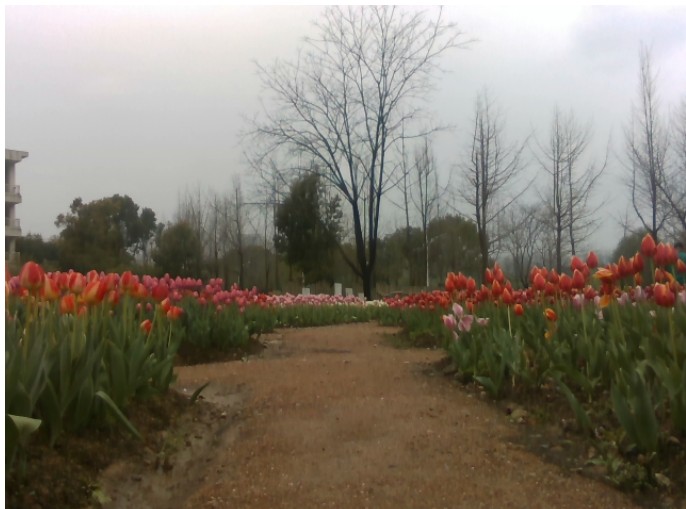

**Figure 2.** Original image.

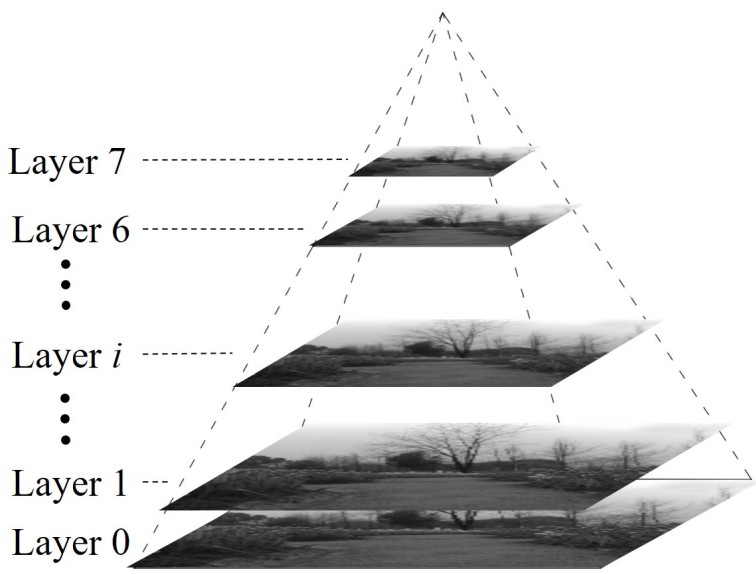

**Figure 3.** Gaussian image pyramid.

According to Mur-Artal et al. [1] and iterative test results of feature point quality at different layer numbers, the total number of layers in the image pyramid is determined to be $m = 8$, and the scaling factor is $s = 1.2$. The number of feature points in each layer of the image pyramid is allocated according to the image area, which is prepared for uniform feature points. If the total number of feature points in the image pyramid is $N = 500$, the number of feature points assigned from layers 0 to 7 is 108, 91, 75, 63, 53, 44, 37, and 29, respectively.

### 2.2.2. Adaptive Threshold $T_a$ Based on the Mesh Region

To better extract the ORB feature points of the image as the points to be matched, the image must be meshed (the mesh size is 30 × 30 pixels), and the adaptive threshold $T_a$ in each mesh must be obtained according to the Michelson contrast $C_M$ of the image. The larger the $C_M$, the more distinctive the texture features of the image in the current mesh [20], and the following equation shows the corresponding relationship between $T_a$ and $C_M$:

$$T_a = K \times C_M \times I_{avg} \tag{1}$$

where $K$ is the proportional coefficient, $0 < K < 1$; $I_{avg}$ is the average gray value of the pixels in the mesh, and the formula for $C_M$ is as follows:

$$C_M = \frac{I_{max} - I_{min}}{I_{max} + I_{min}} \tag{2}$$

where $I_{max}$ and $I_{min}$ are the maximum and minimum grey values of the pixels in the mesh, respectively.

The fixed threshold ($T = 40$) determined by the test results of accuracy and aggregation rate of extracted feature points under different thresholds and the adaptive threshold $T_a$ in this study are used to extract feature points from Figure 2. To better compare the effect of extraction, Figure 2 is subjected to erosion treatment, and the feature points are presented in the form of blue dots as shown in Figure 4.

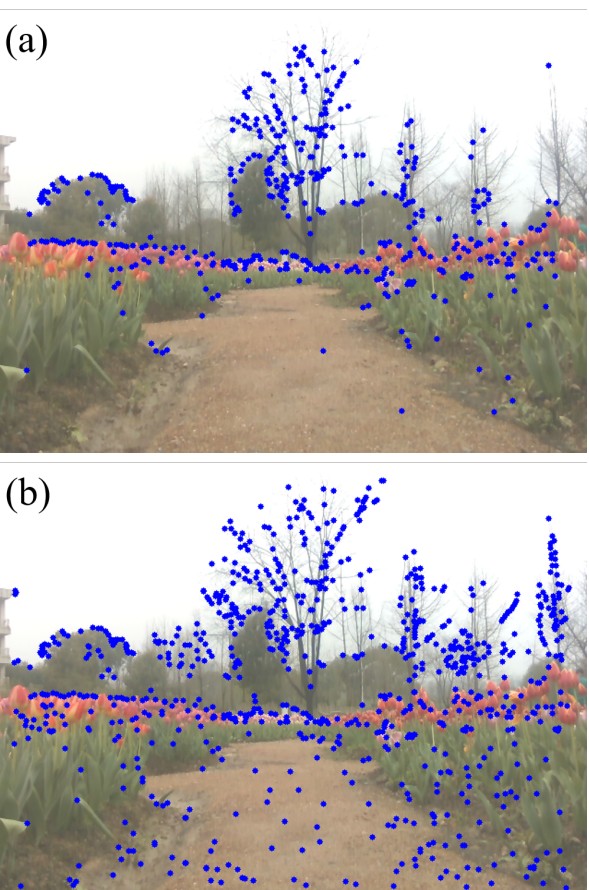

**Figure 4.** Feature point extraction results under different methods: (**a**) Fixed threshold; (**b**) adaptive threshold $T_a$.

The number of feature points in Figure 4a is 439 and that in Figure 4b is 861. It can be seen that this method can make full use of the information of each region of the image

to extract feature points and provide more abundant points to be matched for subsequent image feature matching.

### 2.2.3. Uniform Feature Points Based on Quadtree

It can also be seen from Figure 4b that the feature points extracted by the adaptive threshold Ta still have some problems, such as uneven distribution of feature points and more redundant features. This will lead to a non-negligible error in the interframe position and attitude calculation, which reduces the positioning accuracy. Therefore, the quadtree structure is applied to further uniformize the extracted feature points [21].

As shown in Figure 5, the original image is divided into four subregions (i.e., $n_1$–$n_4$) according to the area. Then the number of feature points contained in each region ($N_p$) is determined; if $N_p > 1$, the region is further divided into four subregions (for example, the $n_1$ region, which contains five feature points, continues to be divided into $n_{1-1}$–$n_{1-4}$); if $N_p = 1$, the area is retained; if $N_p = 0$, this area is deleted. When the total number of regions is greater than the number of feature points to be extracted, or the total number of region division (in Figure 5, the region is divided for a total of three times) is greater than the threshold, no more new areas will be divided, which means that the feature points are uniform.

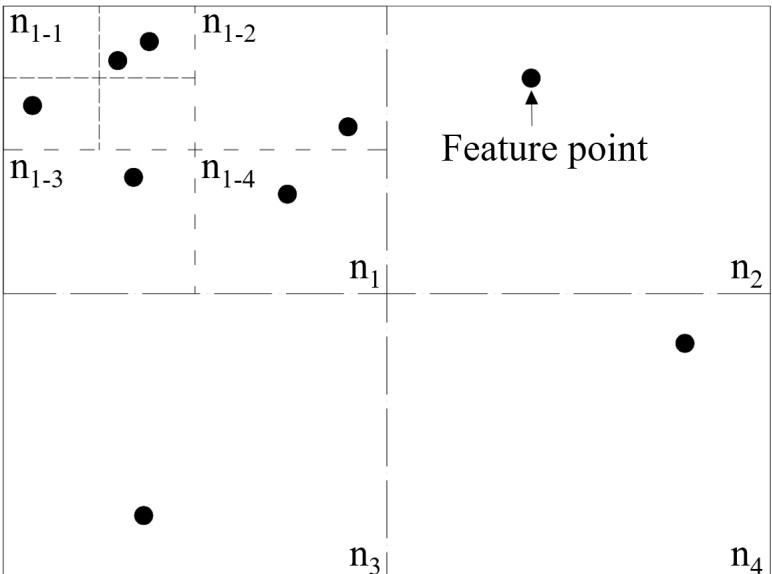

**Figure 5.** Schematic of the quadtree structure.

In an actual division, there may still be multiple feature points in a certain region after uniforming is completed. The Harris operator [22] is used to suppress multiple feature points in the area [23], keeping only the feature points with the most significant Harris response intensity, to make the distribution of feature points more uniform and reduce feature redundancy. The effect of uniforming is shown in Figure 6.

The aggregation rate $c$ is used to evaluate the accuracy of the feature point extraction, which can be expressed as follows:

$$c = \frac{N_c}{N_a} \times 100\% \tag{3}$$

where $N_c$ is the total number of aggregation points (if there are more than three feature points in a certain range near a feature point, then it is defined as the aggregation point), and $N_a$ is the total number of feature points extracted from the image. The closer $c$ is to 0, the better the accuracy of the feature extraction. In addition, the distribution uniformity $\rho$

is used to evaluate the uniformity of feature point extraction, which can be described by the following equation:

$$\rho = \frac{P}{M} \tag{4}$$

where $M$ is the total number of meshes obtained by meshing the image with a mesh size of $30 \times 30$ pixels, and $P$ is the total number of meshes with feature points in the mesh. The closer $\rho$ is to 1, the better the uniformity of the feature point extraction.

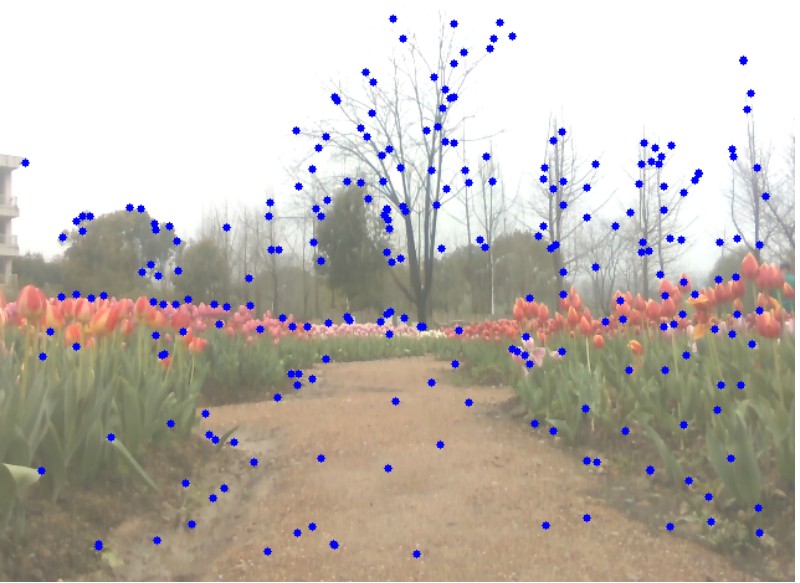

**Figure 6.** Extraction effect of uniform ORB feature points.

The above two indexes were used to quantify the feature point set of Figures 5b and 6, and the results are shown in Table 1.

**Table 1.** Comparison of the aggregation rate and uniformity ($c$ and $\rho$) of the nonuniformed and uniformed feature point set.

| Feature Point Set | Aggregation Rate $c$ (%) | Uniformity $\rho$ |
|---|---|---|
| Nonuniformed | 60.42 | 0.08 |
| Uniformed | 21.42 | 0.44 |

Table 1 indicates that the method proposed in this study can effectively improve the accuracy and uniformity of feature point extraction and eliminate redundant points to be matched in a subsequent study.

### 2.3. Combined Feature Matching

Brute-force matching [24] is widely adopted in ORB feature matching by calculating the Hamming distance [25] between the feature descriptors. However, when the number of points to be matched is large, the time consumed by this method increases, thereby affecting the matching efficiency. Therefore, a combined feature matching algorithm based on the improved LK optical flow and feature descriptor is proposed to improve the efficiency and accuracy of feature matching.

#### 2.3.1. Improved LK Optical Flow Method

In computer vision, optical flow refers to the distance and direction of a pixel moving between images over time and reflects the relationship between the changing information of the image and the motion of the object. The traditional LK optical flow method is based on the assumption of grayscale invariance and uses the brightness difference between

two images to track the instantaneous velocity of feature points [26]. However, when the motion of the two images is large or the brightness changes, the optical flow estimated by this method is inaccurate, and the robustness is poor. In this study, the gradient calculation method of the original algorithm was improved, and the number of self-changing iterations was set according to the estimation of the reversible condition number of the Hessian matrix to improve the robustness and efficiency of the algorithm. The specific concepts are as follows.

The coordinates of the feature point set to be matched are reduced to each image layer according to the image pyramid scaling factor. When calculating the optical flow of a feature point, the calculation starts from the top layer of the images, and then the optical flow result of the previous layer is taken as the initial optical flow of the next layer so that the calculation of the entire optical flow is completed step-by-step from coarse to fine. $\boldsymbol{g}_{i,j}$ is the initial optical flow of the $j$ feature point of the layer $i$ image of the image pyramid, which can be represented as follows:

$$\boldsymbol{g}_{i,j} = \begin{cases} 0, i = n - 1 \\ 2\left(\boldsymbol{g}_{i+1,j} + \boldsymbol{d}_{i+1,j}\right), 0 \le i \le n - 1 \end{cases} \tag{5}$$

where $\boldsymbol{d}_{i,j}$ denotes the residual optical flow. As shown in Figure 7, $\boldsymbol{g}_{i,j}$ determines the initial position of the point to be matched in the matching image, and $\boldsymbol{d}_{i,j}$ estimates the exact position of the matching point based on the assumption of grayscale invariance on the basis of $\boldsymbol{g}_{i,j}$.

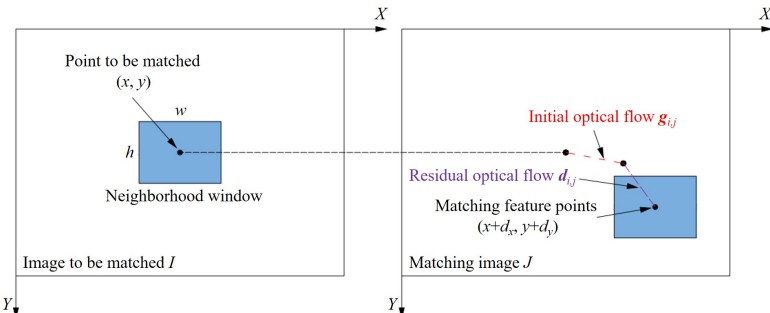

**Figure 7.** Schematic of optical flow.

In this study, $\boldsymbol{d}_{i,j}$ was calculated using iterative optimization. The optimal pixel offset was estimated by minimizing the square difference of the neighborhood window pixel gray values in the image to match $I$ and the matching image $J$. When an iteration satisfies the iterative condition (i.e., the step size of the iteration is less than a certain threshold or the number of iterations is greater than the set value), an accurate residual optical flow $\boldsymbol{d}_{i,j}$ is assumed as having been obtained.

Suppose that the coordinates of the point to be matched are $(x, y)$, the size of the neighborhood window is $w \times h$, and the residual optical flow of the point is $\boldsymbol{d}(d_x, d_y)$. Then, the pixel square difference $E$ of the neighborhood window can be expressed as follows:

$$E(\boldsymbol{d}) = \sum_{x=x-w}^{x+w} \sum_{y=y-h}^{y+h} \left(I(x,y) - J(x + d_x, y + d_y)\right)^2 \tag{6}$$

The partial derivative of $\boldsymbol{d}$ can be obtained from Equation (6):

$$\frac{\partial E}{\partial \boldsymbol{d}} = -2 \sum_{x=x-w}^{x+w} \sum_{y=y-h}^{y+h} \left(I(x,y) - J(x + d_x, y + d_y)\right) \begin{bmatrix} \frac{\partial J}{\partial x} & \frac{\partial J}{\partial y} \end{bmatrix} \tag{7}$$

Because the difference between the frames in the optical flow hypothesis is small, a Taylor series expansion is performed for $J(x + d_x, y + d_y)$ in Equation (7) and the first-order term is retained as follow:

$$J(x + d_x, y + d_y) \approx J(x,y) + \begin{bmatrix} \frac{\partial J}{\partial x} & \frac{\partial J}{\partial y} \end{bmatrix} \boldsymbol{d} \tag{8}$$

Substituting Equation (8) into Equation (7) yields:

$$\frac{\partial E}{\partial d} = -2 \sum_{x=x-w}^{x+w} \sum_{y=y-h}^{y+h} \left( I(x,y) - J(x,y) - \begin{bmatrix} \frac{\partial J}{\partial x} & \frac{\partial J}{\partial y} \end{bmatrix} \boldsymbol{d} \right) \begin{bmatrix} \frac{\partial J}{\partial x} & \frac{\partial J}{\partial y} \end{bmatrix} \tag{9}$$

Suppose:

$$\delta I = I(x,y) - J(x,y) \tag{10}$$

$$\nabla \boldsymbol{I} = \begin{bmatrix} \frac{\partial J}{\partial x} & \frac{\partial J}{\partial y} \end{bmatrix} \approx \begin{bmatrix} \frac{\partial I}{\partial x} & \frac{\partial I}{\partial y} \end{bmatrix} \tag{11}$$

It can be seen from Equation (11) that in the traditional optical flow algorithm, to reduce the calculation time, the gradient of the points on the image to be matched is used instead of the gradient of the points on the matching image. However, when the motion offset between the image to be matched and matching image is large, this approximation method leads to a decrease in the matching effect. In this study, the gradient of the neighborhood window of the matching point was calculated, as shown as follows:

$$\nabla \boldsymbol{I} = \begin{bmatrix} I_x & I_y \end{bmatrix} = \begin{bmatrix} \frac{\partial J}{\partial (x+d_x)} & \frac{\partial J}{\partial (y+d_y)} \end{bmatrix} \tag{12}$$

In addition, suppose:

$$\boldsymbol{G} = \sum_{x=x-w}^{x+w} \sum_{y=y-h}^{y+h} \begin{bmatrix} I_x^2 & I_x I_y \\ I_x I_y & I_y^2 \end{bmatrix} \tag{13}$$

$$\boldsymbol{b} = \sum_{x=x-w}^{x+w} \sum_{y=y-h}^{y+h} \begin{bmatrix} \delta I \cdot I_x \\ \delta I \cdot I_y \end{bmatrix} \tag{14}$$

Substituting Equations (13) and (14) into Equation (9) and making Equation (9) equal to 0, that is, finding the minimum value of Equation (6), the following formula can be obtained:

$$\boldsymbol{d} = \boldsymbol{G}^{-1}\boldsymbol{b} \tag{15}$$

If the matching point is moved along $\boldsymbol{d}$, the residual optical flow at the new position of the matching point is calculated, and iterations are performed until the iteration condition is met, the optimal estimated optical flow can be obtained.

Because the number of iterations has a significant influence on the quality of the final remaining optical flow and the time consumed by the algorithm, the Hessian matrix $\boldsymbol{G}$ is used to set the number of self-changing iterations. The self-changing iteration number takes the Hessian matrix $\boldsymbol{G}$ as an evaluation coefficient. If the change in the evaluation coefficient of an iteration is less than the threshold, the iteration is considered to be over. In addition, the maximum number of iterations and minimum step size are still limited to prevent the condition from failing. The entire iteration ends when the following condition is met:

$$\mathrm{rcond}(\boldsymbol{G}_k) - \mathrm{rcond}(\boldsymbol{G}_{k-1}) < 10^{-5} \ \text{ or } \ \|\boldsymbol{d}\|_2 < 0.03 \tag{16}$$

where $k$ is the current number of iterations, and the function rcond returns the estimation of the condition number of the invertible matrix. By self-changing the number of iterations, when the results are almost the same as the optimal results, the iteration ends by comparing the changes in the iterative evaluation coefficient, reducing the number of iterations, and time required by the algorithm.

### 2.3.2. Feature Rough Matching Based on Improved ORB-LK Optical Flow

The improved LK optical flow proposed in this paper was used to trace the uniform ORB feature points in Figure 6 and to estimate the position of the matching points on the matching image (Figure 8a), thus completing the feature rough matching, as shown in Figure 8b.

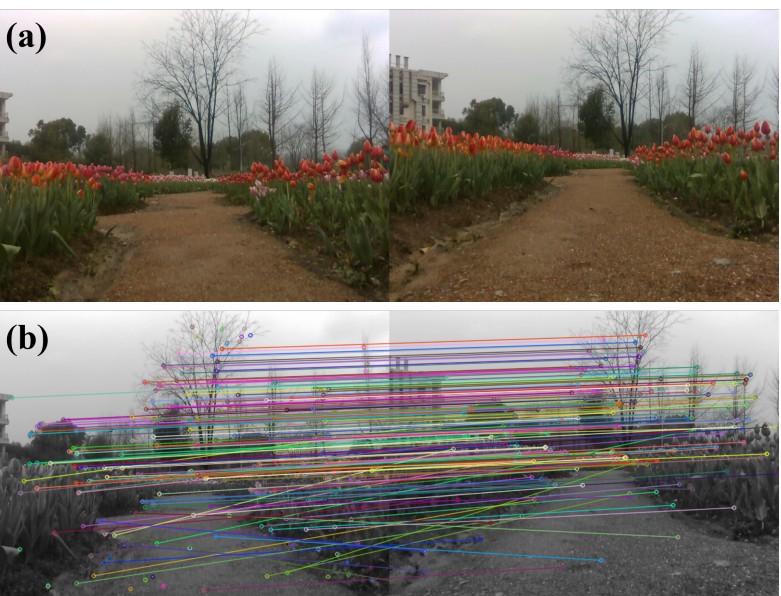

**Figure 8.** Feature rough matching. (**a**) Matching image; (**b**) results of feature matching.

There were 194 matching pairs in Figure 8b, 42 pairs of which were mismatched, and the matching accuracy was only 78.35%, owing to the limitations of the LK optical flow method in estimating the pixel moving optical flow through the pixel grayscale. When the difference between the image to be matched and the matching image is significant, the strong assumption based on grayscale invariance is challenging to satisfy, resulting in a certain number of misestimates.

### 2.3.3. Feature Precise Matching Based on Feature Descriptor

To further eliminate the mismatches in feature rough matching, a precise feature matching method based on feature descriptors is proposed in this paper. The specific concepts are as follows:

First, the direction of the feature points is calculated using the grayscale centroid method [27] for pairs of matching points obtained by rough matching. Second, using the direction information, the rotated descriptor (i.e., Steer BRIEF) is calculated [28]. The descriptor is a one-dimensional vector of size 256, which has elements of 0 or 1, and the binary piecewise function $\tau$ is defined as follows:

$$\tau(I; u, v) = \begin{cases} 1, I(u) < I(v) \\ 0, else \end{cases} \tag{17}$$

where $I(u)$ and $I(v)$ are the gray values of pixels $u$ and $v$ in the image $I$, respectively.

The descriptor vector of the point pair is then used to calculate the Hamming distance between the two vectors to measure the similarity between the two points. Let the feature point descriptor vector to be matched be $V_p$, and let the descriptor vector of the matching point be $V_c$, then the Hamming distance $H$ of the two descriptors is as follows:

$$H = \sqrt{\sum_{d=1}^{256} \left( V_{p,d} - V_{c,d} \right)^2} \tag{18}$$

where $d$ denotes the current dimension of the descriptor vector.

Finally, we consider twice the minimum Hamming distance $H_{min}$ as the threshold $T_{ham}$. If the Hamming distance of the matching point pair is greater than $T_{ham}$, it is considered to be a mismatch and is eliminated. As shown in Figure 9, 99 pairs were obtained using feature precise matching based on feature rough matching, of which only seven pairs were

mismatched, and the matching accuracy was 92.93%, which was 14.58% higher than that of rough matching. This method effectively improves the accuracy of feature matching.

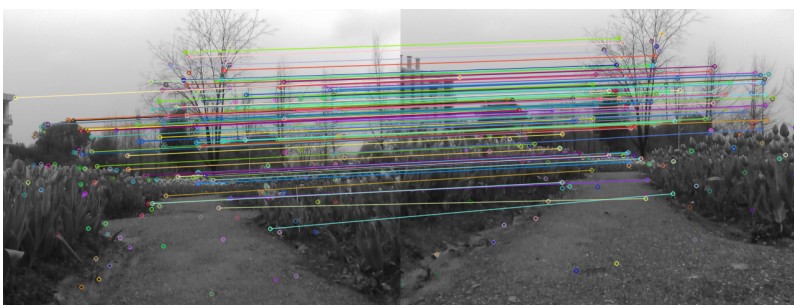

**Figure 9.** Results of feature precise matching.

### 3. Experimental Results and Discussion

To verify the effectiveness of the proposed algorithm, two experiments were designed for feature point extraction and feature matching. In addition, the corresponding existing algorithms were selected to compare with the proposed algorithm in each experiment. The experimental site was selected at the educational teaching practice base of Zhejiang A&F University (119°44′17″N, 30°15′42″E), which mainly grows ornamental flowers such as tulips and lilies, and its satellite image is shown in Figure 10. A four-wheel-independent-drive and steering (4WID-4WIS) mobile platform was used as an experimental platform. An Intel RealSense D435i camera with an image resolution of 640 × 480 pixels was installed on the experimental platform to collect the experimental image data. The experiments were performed on the Ubuntu18.04 operating system. The CPU model of the computer was AMD R7 4800H, with a memory capacity of 16 GB. The experimental setup is shown in Figure 11.

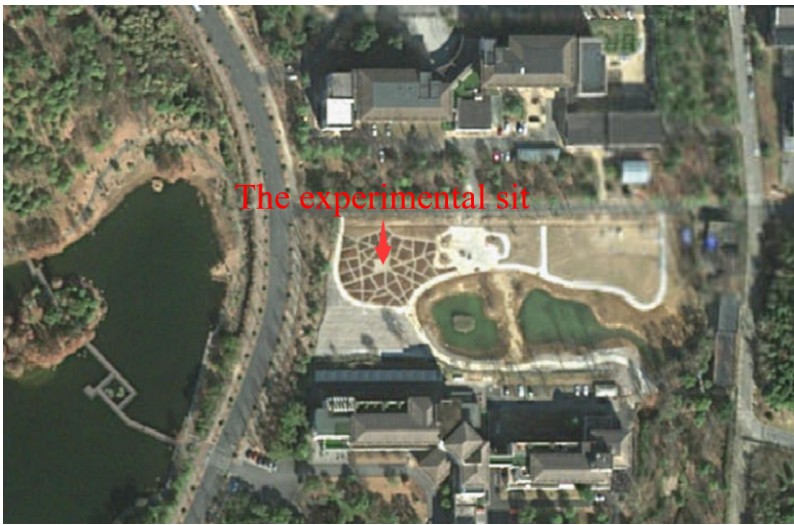

**Figure 10.** The satellite image of the experimental site.

### 3.1. Quality and Analysis of Feature Point Extraction

#### 3.1.1. Results

Four types of typical working scene images of a horticultural robot were collected for the feature extraction experiment: normal light, weak light, high texture, and low texture scenes (Figure 12). In the contrast experiment, the traditional ORB feature extraction algorithm was used, the extraction threshold of this algorithm was set to 30, and the number of feature points was 500. The result of the feature point extraction is shown in Figure 13. In addition, to verify the improved accuracy and uniformity of the ORB feature extraction algorithm proposed in this paper, starting from the time consumption of the algorithm,

the uniformity of the feature point distribution, and the accuracy of feature point extraction, the above two algorithms were tested three times in four scenes, and the average results are shown in Table 2.

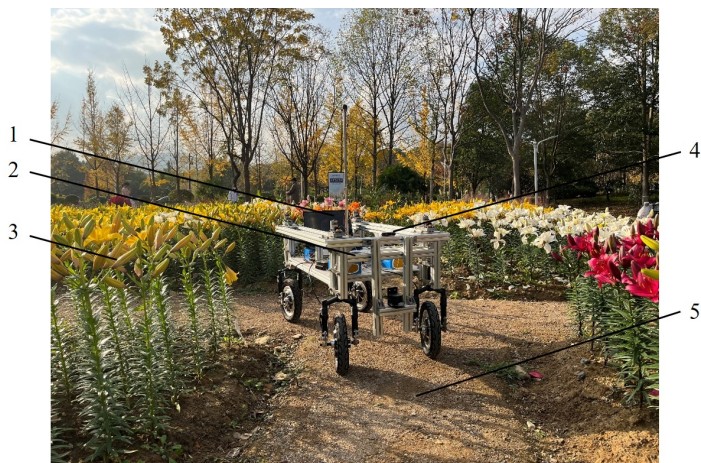

**Figure 11.** Experimental environment. 1. PC; 2. experimental platform; 3. flowers; 4. Intel RealSense D435i; and 5. unstructured path.

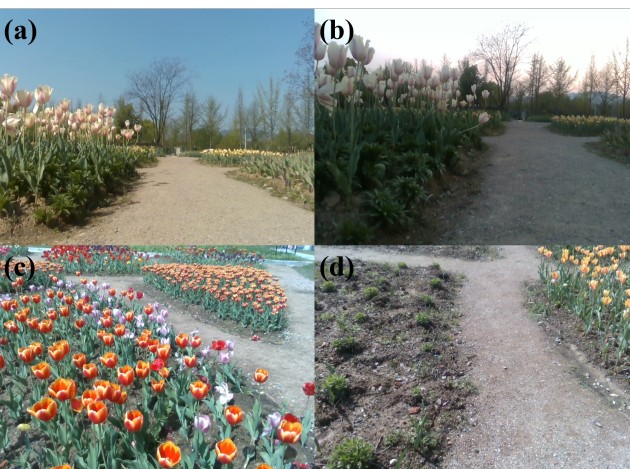

**Figure 12.** Original images from four different scenes: (**a**) Normal light scene; (**b**) weak light scene; (**c**) high texture scene; (**d**) low texture scene.

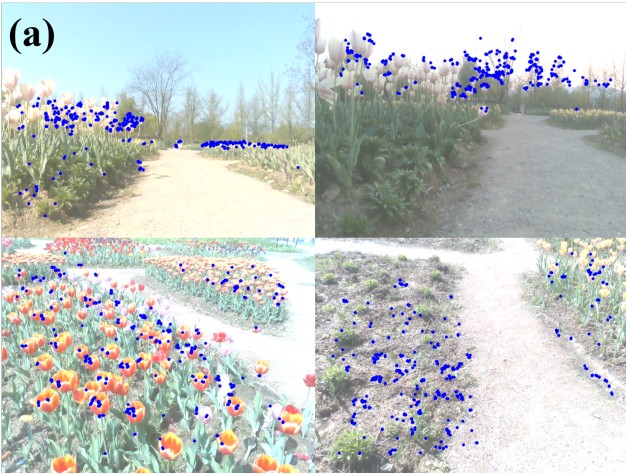

**Figure 13.** *Cont.*

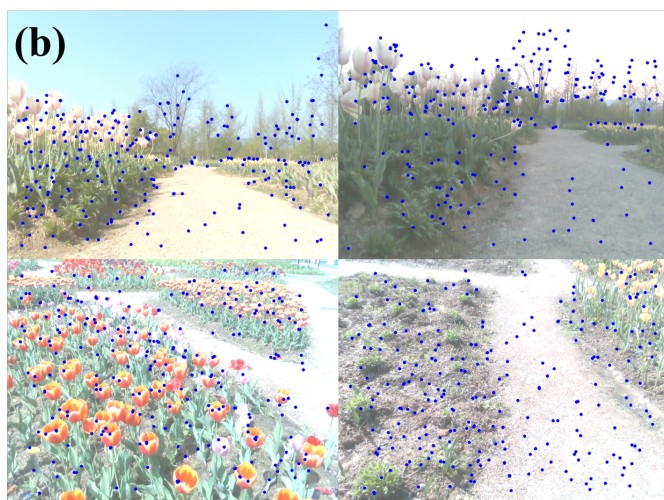

**Figure 13.** Comparison of feature extraction results of the two algorithms. (**a**) Traditional ORB algorithm; (**b**) proposed algorithm.

**Table 2.** Comparison of algorithm data indicators in different scenes: $N_f$ is the number of feature points; $t$ is the time consumption of the algorithm; $\rho$ is the uniformity; and $c$ is the aggregation rate.

| Scene | Traditional ORB Algorithm | | | | Proposed Algorithm | | | |
|---|---|---|---|---|---|---|---|---|
| | $N_f$ | $t$/ms | $\rho$ | $c$/% | $N_f$ | $t$/ms | $\rho$ | $c$/% |
| Normal light | 500 | 7.74 | 0.15 | 74.80 | 251 | 3.72 | 0.40 | 23.36 |
| Weak light | 500 | 7.34 | 0.16 | 69.43 | 250 | 3.13 | 0.40 | 12.02 |
| High texture | 500 | 12.53 | 0.29 | 61.46 | 249 | 7.22 | 0.47 | 14.28 |
| low texture | 500 | 11.92 | 0.31 | 57.39 | 250 | 5.03 | 0.51 | 11.56 |

3.1.2. Discussion

According to Figure 13, most of the feature points extracted by the traditional ORB algorithm are concentrated in areas with a high edge texture in the image. However, in areas with weak textures, such as roads, the ability of this algorithm to extract feature points is weak. Therefore, the feature points obtained using this method can not sufficiently reflect the overall changes in the image. By contrast, the feature points extracted by the improved ORB extraction algorithm proposed in this study had a good distribution in the entire image, and the extraction effect was less affected by the change in illumination.

Table 2 indicates that, compared with the traditional ORB algorithm, the algorithm proposed in this paper improved the uniformity by an average of 0.22 and reduced the aggregation rate by 50.47% on average, so it had better uniformity and accuracy. In addition, the average time consumption of this algorithm was 4.78 ms, which was 5.10 ms shorter than the 9.88 ms of the traditional ORB algorithm, which improved the efficiency of feature extraction. Simultaneously, the algorithm in this study eliminated many redundant feature points, which reduced the amount of computation required for feature matching.

*3.2. Accuracy and Analysis of Feature Matching*

3.2.1. Results

Images from four scenes (Figure 14) were collected for the feature matching experiment. The comparison algorithm used the brute force (BF) and LK optical flow methods. For example, the matching results of a normal light scene obtained using the three algorithms are shown in Figure 15. Experiments were carried out three times in four scenes using three algorithms, and the average values of the matching number, matching time, total time, and matching accuracy were calculated. The results are shown in Figure 16. In addition, Figure 17 shows the effect of the feature matching of images in the other three scenes using the proposed algorithm.

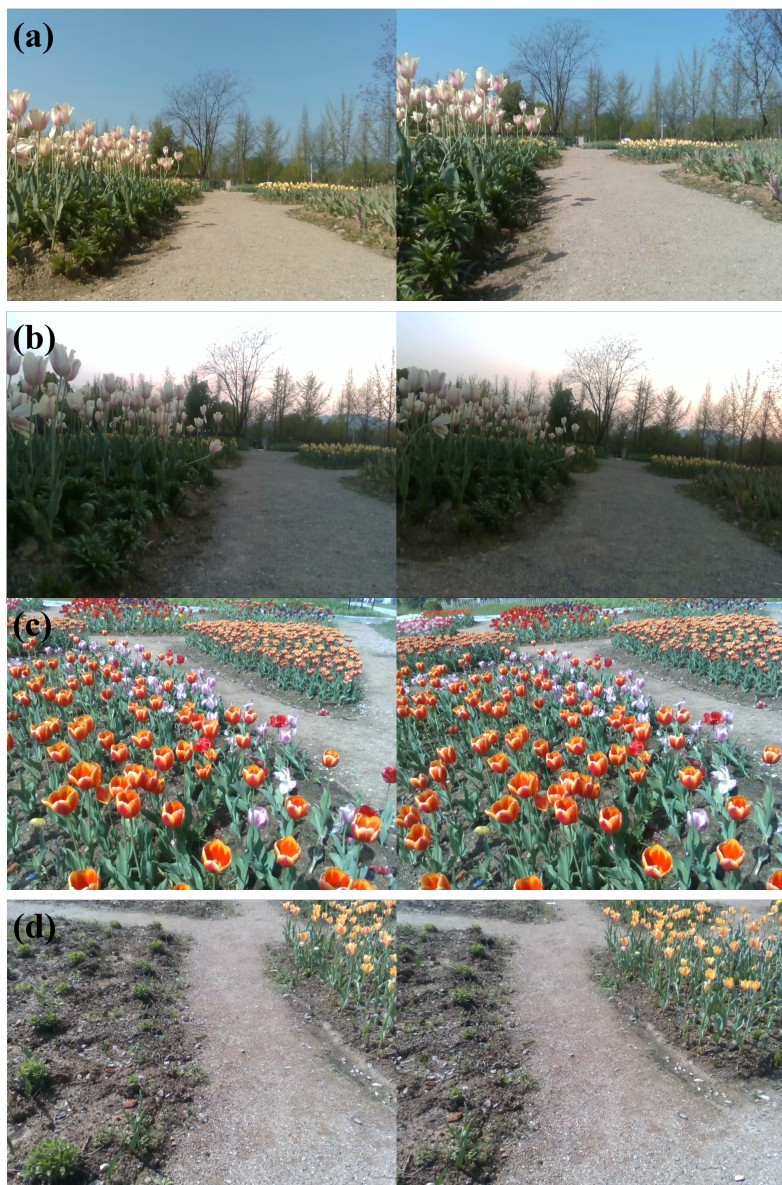

**Figure 14.** Experimental images under different scenes: (**a**) Normal light scene; (**b**) weak light scene; (**c**) high texture scene; (**d**) low texture scene.

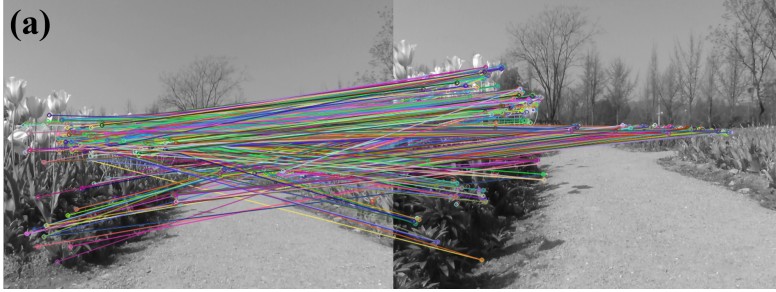

**Figure 15.** *Cont.*

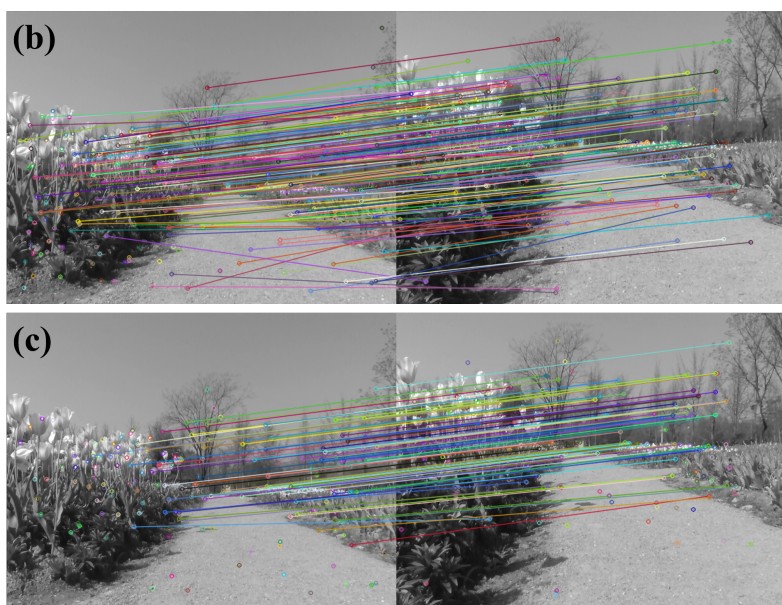

**Figure 15.** Experimental results of a normal light scene under different algorithms: (**a**) BF matching; (**b**) LK optical flow; (**c**) proposed algorithm.

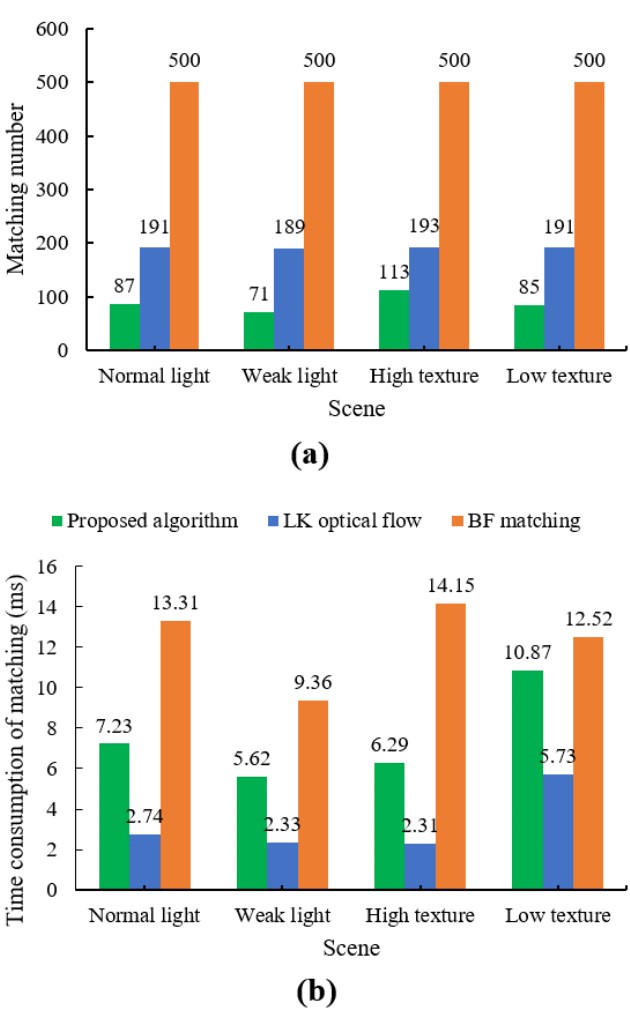

**Figure 16.** *Cont.*

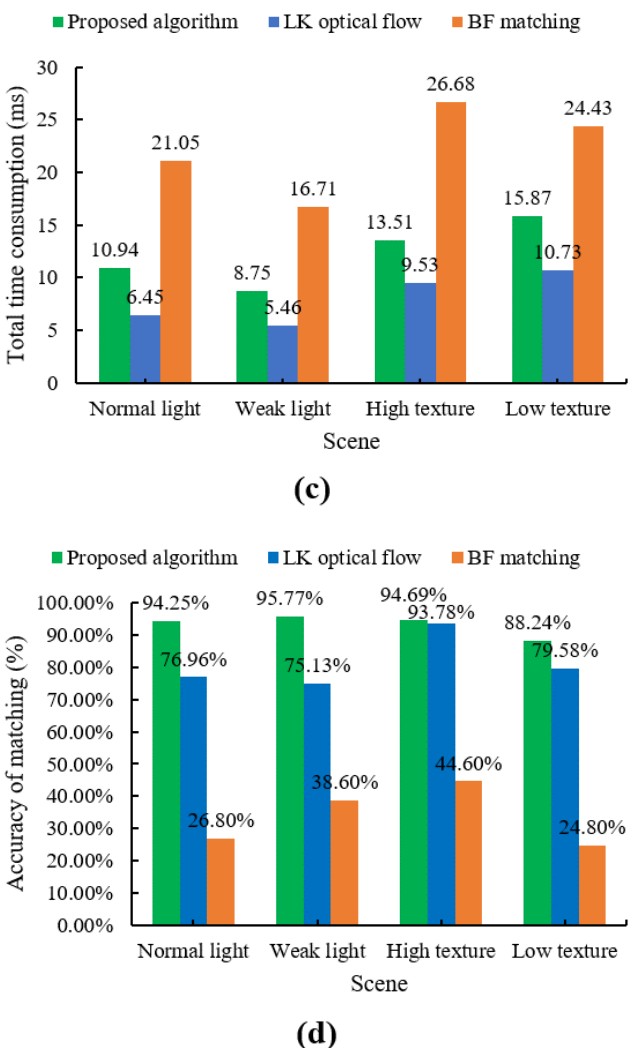

**Figure 16.** Statistical results of matching under three algorithms: (**a**) matching number; (**b**) time consumption of matching; (**c**) total time consumption; (**d**) accuracy of matching.

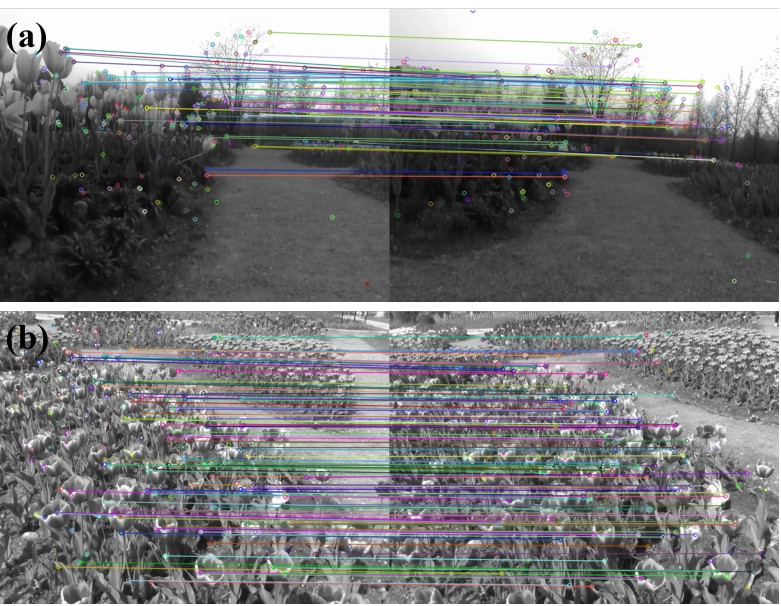

**Figure 17.** *Cont.*

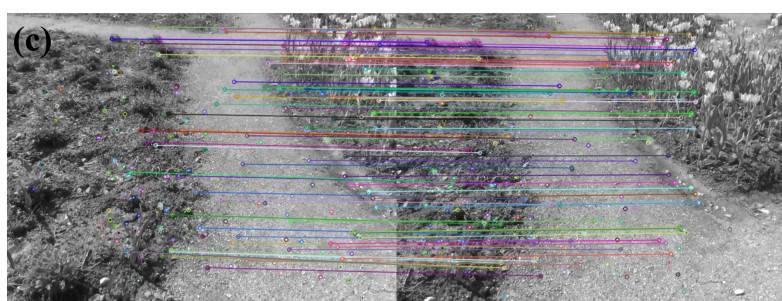

**Figure 17.** Results of feature matching using the proposed algorithm under three other scenes: (**a**) weak light scene; (**b**) high texture scene; and (**c**) low texture scene.

### 3.2.2. Discussion

As shown in Figures 15 and 16a, the algorithm proposed in this study removes redundant feature points through improved feature extraction, thus reducing invalid feature matching. In addition, this algorithm uses feature precise matching to eliminate mismatching on the basis of improved LK optical flow. Although it was inferior to, the traditional BF matching method and LK optical flow method with regard to matching numbers, the distribution of feature matching was more uniform and could better cover the change information of the entire image.

According to Figure 16b,c, the average time consumption of matching and total time consumption of the proposed algorithm were 7.50 and 12.27 ms, respectively, 39.18% and 44.79% shorter than those of the traditional BF matching algorithm, and the real-time performance was greatly improved. However, to achieve more accurate feature matching, the time consumption of feature matching and the total time consumption of feature extraction and feature matching were slightly higher than those of the LK optical flow method.

From Figure 16d, the matching accuracy of the proposed algorithm was 93.24% on average while that of LK optical flow was 81.37%. Thus, matching accuracy was improved by 14.59% on average. Moreover, the difference between the highest matching accuracy and the lowest matching accuracy of the LK optical flow and the BF matching method in the four scenes was 18.65% and 19.8%, respectively, while the proposed algorithm was only 7.53%. the accuracy performance in various scenes was more stable and robust than that of the other two algorithms.

Figure 17 indicates that the algorithm in this study could obtain a good matching quality in the other three scenes, which shows that it can adapt to image matching tasks in various scenes.

### 4. Conclusions

In this paper, a novel horticultural image feature matching algorithm based on improved ORB and LK optical flow is proposed. The proposed algorithm combines the high accuracy of the feature point method and the high efficiency of the LK optical flow method, and exhibits good robustness in various horticultural environments. The comparison results reveal that this algorithm improves the uniformity and accuracy of feature point extraction by 0.22 and 50.47%, respectively. In addition, in comparison to the LK optical flow method, this algorithm has a 14.59% higher accuracy with regard to feature matching, and the average matching time consumption and total time consumption are lower by 39.18% and 44.79%, respectively. In different scenes, the average of feature matching accuracy obtained by the proposed algorithm can reach 93.24%. These results make the algorithm suitable for use in the V-SLAM process of horticultural robots, where it could improve the accuracy of the robot's real-time positioning and map construction. In addition, this study shows great potential for applications in the fields of target recognition in industrial logistics and image stitching for pest and disease detection.

**Author Contributions:** Conceptualization, L.Y.; Data curation, Q.C., Y.L. and Y.Y. (Yankun Yang); Formal analysis, Q.C.; Funding acquisition, L.Y.; Investigation, Q.C. and Y.Y. (Yuncong Yang); Methodology, Q.C. and T.X.; Supervision, L.X.; Software, Q.C.; Validation, L.Y. and L.X.; Visualization, L.Y. and L.X.; Writing—original draft, Q.C. and L.Y.; Writing—review and editing, Q.C., L.X. and L.Y. All authors have read and agreed to the published version of the manuscript.

**Funding:** This research was funded by the Key R&D Program of Zhejiang (2022C02042) and National Undergraduate innovation training program (202101341050).

**Data Availability Statement:** The data presented in this study are available on request from the corresponding author. The data are not publicly available due to another study related to this data is not yet publicly available.

**Acknowledgments:** We thank the editors for reviewing the manuscript, and the anonymous reviewers for providing suggestions that greatly improved the quality of the work.

**Conflicts of Interest:** The authors declare no conflict of interest.

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
