# Peer review of "Horticultural Image Feature Matching Algorithm Based on Improved ORB and LK Optical Flow"

_remotesensing, doi:10.3390/rs14184465_

Round 1

Reviewer 1 Report

A novel horticultural image feature matching algorithm based on improved ORB and LK optical flow is proposed. The proposed algorithm combines the high accuracy of the feature point method and the high efficiency of the LK optical flow method, and exhibits good robustness in various horticultural environments.

1. The description of the present method is not clear. Figure 1 must be redrawn with a clear expression of the method.

2. What is the contribution? Not clear

3. Compare the present method with existing ones.

Author Response

Dear reviewer,

thank you for your helpful suggestions and kindly processing. All comments have been carefully considered for revision and the main modified parts have been highlighted in red in the revised version. Each comment is also responded one by one on the attachment. Please see the attachment.

Reviewer 2 Report

The paper is well-written and focuses on an interesting topic. The authors proposed a new approach for image feature matching based on improved ORB and LK optical flow. They carried out several experiments and compared their algorithm results with 2 other approaches (BF matching and LK optical flow). Their algorithm achieved better results while being a little bit more time-consuming than the LK optical flow. From my point of view, I would like to suggest some changes in the text. At the beginning of Section 3, add more details concerning the experimental platform and the experimental setup (why they did not use RTK-GNSS data to obtain a ground truth?). In Section "3.2.2. Discussion" describe in detail the figures 13 to 16. They did it briefly, but a have the feeling that they could explore more the figures and help the readers to better understand their experiments. Finally, in the conclusions section, Section 4., they affirm that the results prove the algorithm is suitable for use in the V-SLAM process of horticultural robots. Why only this kind of application?  

Author Response

(The authors gave the same response as above.)

Reviewer 3 Report

1) The first line of the abstract talks about low accuracy. However, the specific characteristic features of the proposed method which enhances the accuracy (I am not sure) is not mentioned.

2) Are the same dataset used in other works for comparison? How do you validate the improvement in the accuracy mentioned in the abstract?

3) Several critical comments have been given on other people's work. Are all of them solved in this work?

4) How do you define the number of layers in section 2.2.1?

5) How do you fix the threshold in section 2.2.2?

6) How do you validate that the extracted features have enhanced the accuracy? (other than the experimental results based validation)

7) The success of the features are proved mostly based on the classification results. Is it possible to give such an analysis.

8) There is an opinion that the experimental results are less in quantum

Author Response

(The authors gave the same response as above.)

Round 2

Reviewer 1 Report

It can be accepted now.

Reviewer 3 Report

It can be accepted now